# The Community Assessment of Psychic Experiences-Positive scale (CAPE-P15) accurately classifies and differentiates psychotic experience levels in adolescents from the general population

D. Núñez[1,2,3]*, M. I. Godoy[4], J. Gaete[2,5], M. J. Faúndez[1,2], S. Campos[1,2], A. Fresno[1,3], R. Spencer[1,3]

1 Faculty of Psychology, Universidad de Talca, Talca, Chile, 2 ANID, Millennium Science Initiative Program, Millennium Nucleus to Improve the Mental Health of Adolescents and Youths, Imhay, Chile, 3 Faculty of Psychology, Associative Research Program, Center or Cognitive Sciences, Universidad de Talca, Talca, Chile, 4 Departamento de Evaluación, Medición y Registro Educacional, Universidad de Chile, Santiago, Chile, 5 Faculty of Education, Universidad de Los Andes, Santiago, Chile

* dnunez@utalca.cl

**Data Availability Statement:** All relevant data are within the paper and its Supporting information files.

## Abstract

### Background

There is increasing interest in studying psychotic symptoms in non-clinical populations, with the Community Assessment of Psychic Experiences-Positive scale (CAPE-P15) being one of the self-screening questionnaires used most commonly for this purpose. Further research is needed to evaluate the ability of the scale to accurately identify and classify positive psychotic experiences (PE) in the general population.

### Aim

To provide psychometric evidence about the accuracy of the CAPE-P15 for detecting PE in a sample of Chilean adolescents from the general population and classifying them according to their PE severity levels.

### Method

We administered the CAPE-P15 to a general sample of 1594 students aged 12 to 19. Based on Item Response Theory (IRT), we tested the accuracy of the instrument using two main parameters: difficulty and discrimination power of the 15 items.

### Results

We found that the scale provides very accurate information about PE, particularly for high PE levels. The items with the highest capability to determine the presence of the latent trait were those assessing perceptual anomalies (auditory and visual hallucinations), bizarre

**Funding:** This work was supported by: Fondo de Innovación y Competitividad [FIC_40.001.103-0] ANID – Millennium Science Initiative Program [NCS17_035]. https://www.imhay.org/ Programa de Investigación Asociativa (PIA) en Ciencias Cognitivas, Facultad de Psicología, Universidad de Talca [RU-158-2019]. The funders had no role in study design, data collection and analysis, decision to publish, or preparation of the manuscript.

**Competing interests:** The authors have declared that no competing interests exist.

experiences (a double has taken the place of others; being controlled by external forces), and persecutory ideation (conspiracy against me).

## Conclusions

The CAPE-P15 is an accurate and suitable tool to screen PE and to accurately classify and differentiate PE levels in adolescents from the general population. Further research is needed to better understand how maladaptive psychological mechanisms influence relationships between PE and suicidal ideation (SI) in the general population.

## Introduction

Timely detection of psychotic experiences (PE) has been recommended for mental health prevention [1], particularly among adolescents and youths [2]. However, the current capability to identify psychotic manifestations in clinical and general populations is sub-optimal [3]. For instance, 50–62% of patients with psychotic symptoms are not identified in the first contact with mental healthcare services [4], and only 5.2% of cases of emerging first-episode psychosis are detected in secondary mental health services [5]. PE are associated with a wide range of psychiatric symptoms [6,7] and negative outcomes, including poor functioning [8–10], increased use of mental health services and psychotropic medication [11,12], poor treatment response [13,14], and suicidal behavior [15,16]. Consequently, PE are potential markers to identify individuals at risk for psychopathology beyond psychotic spectrum disorders [17,18]. Using short self-report questionnaires could enhance the detection of these individuals [19,20] in the context of stratified programs or sequential testing methods [21,22]. However, the evidence about psychometric properties of available brief measures is still insufficient [22,23].

Over the last years, the focus of psychosis research has increasingly shifted to non-clinical populations [24], as authors have argued that systematic and standardized screening for PE could be feasible in community and educational settings [25–27]. One of the most frequently used screening tools is the Community Assessment of Psychic Experiences (CAPE) [28], which was created following the theoretical framework of the extended subclinical psychosis phenotype [24]. The original version (CAPE-42) [29] has good discriminant validity and good test-retest reliability [19,30]. A shorter version comprises 15 items assessing persecutory ideation (PI), bizarre experiences (BE), and perceptual anomalies (PA) (CAPE-P15) [31]. Evidence for this three-factor structure has been reported in prior studies [32,33], but we also found a strong general factor in a bifactor model among adolescents [34], a finding that requires additional research [33]. The scale has shown good validity and reliability in university students [32]. Additionally, in college students, it has adequate construct and concurrent validity, internal consistency, test-retest reliability, and measurement invariance across sex [35].

The CAPE-P15 scale can help to identify at-risk populations and the screening methodology may consider several approaches. For example, Bukenaite et al. [3], computing the average of all items, identified a cut-off of 1.47 for both frequency and distress scales as suitable for detecting ultra-high risk for psychosis, supporting the CAPE-P15 as a valid and reliable instrument to sensitively and specifically detect positive individuals in adolescent outpatients. Alternatively, it would be possible to establish several risk groups according to the severity of the latent traits using scores derived from Item Response Theory analysis. No previous studies have used the latter approach to assess the accuracy of the CAPE in educational settings, where higher thresholds will probably be needed to reduce false positives [20]. In line with literature

suggesting authors to evaluate the psychometric properties of screening tools for assessing PE [30] and encouraging them to detect PE in both adolescence [6] and population-based samples [22,26,36], we seek to provide new psychometric evidence about the CAPE-P15 in adolescent school students in Chile. First, we tested its internal structure by conducting confirmatory analyses. Second, we assessed its accuracy in classifying PE levels using Item Response Theory (IRT) [37], and tested which items better discriminate the latent trait. Furthermore, we assessed the instrument's discriminant validity by examining associations between PE and symptoms (depression and anxiety), suicidal ideation (SI), and some maladaptive psychological mechanisms (defeat, entrapment, and rumination) probably influencing the experience of psychotic symptoms [38,39] and underlying the links between PE and SI in psychosis [40,41].

## Materials and methods

### Participants

We conducted a cross-sectional study with 1599 adolescents recruited between April and September 2019 in 11 public secondary public schools in Chile. The inclusion criteria were that the students and their parents voluntarily agreed to participate in the study and signed written and informed consent. We excluded five individuals who wrongly defined their ages (out of the age range of 12–19 years). We performed the analyses with a final sample of 1594 adolescents (mean age = 15.56 SD = 1.35, women = 47.4%).

### Measures

**Psychotic experiences.**   We used the CAPE-P15, a 15-item self-report questionnaire [31]. In the current version, responses to items range from 1 (never) to 5 (very often). The scale assesses three domains: paranoid ideation (PI, 5 items), bizarre experiences (BE, 7 items), and perceptual anomalies (PA, 3 items). Scores can range from 15 to 75. Higher scores indicate higher severity of PE. All items are averaged for an overall measure of the trait being assessed. We did not address the degree of distress associated with positive symptoms.

**Depressive symptoms.**   We used the Patient Health Questionnaire-9 (PHQ-9) [42], a 9-itemself-report questionnaire with responses ranging from 0 (not at all) to 3 (nearly every day). Total scores can range from 0 to 27. Scores of 0–4 indicate no depressive symptoms, 5–9 mild severe depressive symptoms, 10–14 moderate depressive symptoms, 15–19 moderately-severe depressive symptoms, and 20–27 severe depressive symptoms [42]. In outpatient adolescents, its positive predictive value was 77% [43]. In Chilean adolescents, Borghero et al. [44] observed the following values: internal consistency = .78; sensitivity = 86.2%, specificity = 82.9%). In our sample, Cronbach's alpha ($\alpha$) was 0.90, and McDonald's Omega Coefficient ($\omega$) was 0.81.

**Anxiety symptoms.**   We used the Generalized Anxiety Disorder 7-item scale (GAD-7) [45], a 7-item self-report questionnaire with possible responses ranging from 0 (not at all) to 3 (nearly every day). Total scores can range from 0 to 21 points. Cutoff scores of5, 10, and 15 respectively represent mild, moderate, and severe levels of anxiety [45]. For scores $\geq$ 11, the scale has shown good predictive value (>99%) and negative value (.83) in adolescents. For scores $\geq$ 17, it was associated with a positive value of.266 and a negative predictive value of >99% [46]. In adolescents from Chile, internal consistency ($\alpha$) reached.86. In our sample, Cronbach's $\alpha$ was 0.90 and McDonald's $\omega$ was 0.86.

**Suicidal ideation (SI).**   We used seven items of the Columbia Suicide Severity Rating Scale (C-SSRS) [47], adapted for use as a self-report questionnaire [48]. The severity of SI was rated on a 7-point ordinal scale in which 1 = wish to be dead, 2 = nonspecific active suicidal thoughts, 3 = thoughts about how to commit suicide, 4 = suicidal thoughts and intentions,

5 = suicidal thought with a detailed plan, 6 = intentions to conduct plan, 7 = prior behaviors or planning acts to commit suicide. Scores over 3 points represent an elevated risk of suicidal ideation. The frequency of SI was addressed by asking participants when these thoughts happened: ever in life ($SI_L$) and/or during the last month ($SI_M$). We only reported the former ($SI_L$) because there were few reports of $SI_M$. In our sample, α was 0.90 and ω was 0.81.

**Defeat and entrapment.**   We used the Short Defeat and Entrapment Scale (SDES) [49]. It comprises eight items with a 5-point response scale ranging from 1 (never) to 5 (very often). Four items assess defeat, defined as the perception of a failed struggle, feelings of powerlessness, and a sense of losing social status or missing personal goals [50]. Four items assess entrapment, defined as the feelings of being threatened or involved in a stressful, unpleasant state or situation which one cannot escape because of internal or external circumstances [51]. Total scores for each scale range from 4 to 20. The scale has shown good reliability (α values from 0.85 to 0.88 (Defeat scale) and from 0.65 to.083 (Entrapment scale) [52]. In our sample, the α and ω values for the Defeat scale were 0.93 and 0.89 respectively, while for the Entrapment scale they were 0.86 and 0.84.

**Rumination.**   We used four items extracted from the Ruminative Response Scale (RRS) [53]. The items assess rumination on a 5-point response scale ranging from 1 (never) to 5 (very often). Total scores range from 4 to 20. In our sample, internal consistency was good (α = 0.87; ω = 0.85).

## Procedure

We invited 11 public schools to participate in the study and all of them agreed to do so after meetings were held with their administration teams. After securing each school's approval and once written and informed consent was obtained from both the adolescents and their caregivers, the participants completed online questionnaires, administered in school computer laboratories. Ethical approval was obtained from the Bioethics Committee of the University of Talca (40.001.103–0; 11/07/2020).

## Data analysis

**Descriptive statistics.**   The descriptive statistics of the participants are reported with percentages and their confidence interval (CI 95%). We used means and standard deviation to describe depressive and anxiety symptoms, suicidal ideation, defeat, entrapment, and rumination. The items of the CAPE-P15 were described by mean, standard deviation, median, skewness and kurtosis parameters, and the quartiles 1 and 3. The two latter parameters are presented in intervals [Q1-Q3], as a robust measure of dispersion. Also, we reported the frequencies of each response category, ranging from 1 (never) to 5 (very often).

**Internal consistency.**   We tested the dimensionality of the CAPE through confirmatory factor analysis (CFA) with weighted least squares (WLS) adjusted over a polychoric matrix, a robust estimator for item ordinals. We used several goodness-of-fit indices which were evaluated when the adjustment was at least acceptable, as described in Table 3. Additionally, we assessed the reliability of the instrument through the omega coefficient (ω). Reliability values of 0.65 or more were regarded as acceptable [54].

**IRT model.**   We used Item Response Theory (IRT) [55] to provide evidence on the capability of the CAPE-P15 and to differentiate participants according to the severity level of their PE. We used two main parameters: discrimination power and difficulty of each item. Moreover, we assessed the accuracy of the scale by computing the item and the test information functions. We categorized the answers as follows: 0 = "never"; 1 = "rarely"; and 2 = "occasionally", "often", or "very often". To assess item discrimination capability, we computed alpha

values, which represent the degree to which the answer categories differentiate between the trait levels and it remains constant for all the thresholds of the categories of the same item. To estimate item difficulty, we used beta parameters ($\beta_1$ and $\beta_2$). $\beta_1$ represents the latent trait that is needed for the adolescent to pass the threshold from answering 0 (never) to 1 (rarely). In other words, it refers to the minimum value of the trait needed to obtain a probability higher than 0.5 of answering option 1. $\beta_2$ represents the threshold for passing from answer category 1 (rarely) to 2 (at least occasionally) (we interpreted these parameters according to Baker et al. [56]).

To evaluate the measurement accuracy of each item in different levels of the latent construct, we tested the trait range θ using the Item Information Function (IFF). Higher information values indicate higher precision of the elements for a certain range of theta.

To assess the overall test and how well it estimates PE severity, we computed the Test Information Function (TIF), which combines the information of all items. Higher information values indicate greater scale precision.

**Relationship of PE with other variables.**   To evaluate associations between PE and other variables, we divided the latent variable estimated by the graded response model into three groups (Low $\leq$ -1 SD; Medium = between -1 and 1 SD; and High $\geq$ 1 SD). Then, we compared PE scores to those of depressive and anxiety symptoms, suicidal ideation, rumination, defeat, and entrapment. For each comparison, we used the median and non-parametric statistic tests for independent groups (Kruskall-Wallis and U Mann-Whitney).

We performed all statistical analyses in R 4.2.0. For the factor analysis, we used the lavaan library, and for the graduated response model, we used the mirt library.

# Results

## Descriptive statistics

Table 1 describes the sociodemographic and clinical characteristics of the participants. Table 2 shows the descriptive scores of the items of the CAPE-P15. We observed asymmetric responses (bias to the left) and a high degree of kurtosis, meaning that the data is concentrated close to the mean. Overall, most participants tend to choose the category "never", except for items 1 and 2, which exhibit greater variability.

## Internal consistency and dimensionality

The fit indices of the CAPE-P15 are good (Table 3). The unidimensional structure of the CAPE (Estimator 1 Factor = 1 general factor reflecting an average score of all items) was corroborated. The hierarchical structure (Estimator Model 2 = 1 general factor plus three correlated factors) was also corroborated. The RMESA index was acceptable for both structures but slightly better for the hierarchical structure (S1 Fig). Thus, the data support the existence of a general PE latent factor, but at the same time, it is possible to differentiate three specific PE dimensions which may have different clinical meanings.

Additionally, the scale has good internal reliability (α = 0.94; ω = 0.81).

## IRT model

**Discrimination and difficulty parameters.**   Table 4 shows parameters α, $\beta_1$, and $\beta_2$. Alpha values vary between 1.05 (moderate) and 2.44 (very high). According to Baker et al. [56], all items have acceptable discrimination capability and properly represent PE as the latent trait in the present sample. We observed moderate values for items BE1 (electronic devices can influence your thoughts) and PA3 (visual hallucinations), high values for items assessing

**Table 1. Demographic variables and measures.**

| Variable | N | Percentage (%) or Mean | [95% Confidence interval] or (SD) |
|---|---|---|---|
| Age | | | |
| 13 years or less | 59 | 3.7 | [2.9–4.8] |
| 14 | 329 | 20.6 | [18.7–22.7] |
| 15 | 404 | 25.3 | [23.2–27.6] |
| 16 | 386 | 24.2 | [22.1–26.4] |
| 17 | 303 | 19.0 | [17.1–21.0] |
| 18 years or more | 113 | 7.1 | [5.9–8.5] |
| Gender | | | |
| Female | 754 | 47.4 | [44.9–49.9] |
| Male | 836 | 52.6 | [50.1–55.1] |
| Repeated grade | | | |
| No | 1,258 | 78.9 | [76.8–80.9] |
| Yes | 336 | 21.1 | [19.1–23.2] |
| Prior psychological treatment | | | |
| No | 1,014 | 63.6 | [61.2–66.0] |
| Yes | 580 | 36.4 | [34.0–38.8] |
| Measures | | | |
| Depressive symptoms | 1.594 | 8.23 | (6.00) |
| Anxiety symptoms | 1.591 | 8.07 | (5.23) |
| Suicidal ideation | 1.594 | 1.19 | (1.88) |
| Defeat | 1.594 | 7.64 | (3.96) |
| Entrapment | 1.594 | 7.29 | (4.00) |
| Rumination | 1.591 | 9.45 | (4.39) |

**Table 2. Descriptive scores of items, CAPE-P15 scale.**

| | Median | [Q1-Q3] | Mean (SD) | Kurtosis | Skewness | Prevalence (%) | | | | |
|---|---|---|---|---|---|---|---|---|---|---|
| | | | | | | 0 | 1 | 2 | 3 | 4 |
| PI1_drop hints | 1 | [0–1] | 1.02 (1.01) | 3.93 | 1.07 | 34.6 | 41.2 | 15.5 | 5.0 | 3.5 |
| PI2_ seem to be | 2 | [1–3] | 1.85 (1.24) | 2.06 | 0.24 | 14.0 | 30.2 | 25.0 | 18.1 | 12.6 |
| PI3_persecuted | 0 | [0–1] | 0.59 (.87) | 5.95 | 1.71 | 59.7 | 28.0 | 7.8 | 3.0 | 1.4 |
| PI4_conspiracy | 0 | [0–1] | 0.49 (.85) | 6.99 | 2.01 | 67.2 | 22.3 | 6.0 | 3.2 | 1.3 |
| PI5_look oddly | 0 | [0–1] | 0.81 (1.07) | 4.40 | 1.41 | 51.2 | 28.9 | 11.4 | 4.5 | 4.1 |
| BE1_electronic devices | 1 | [0–1] | 0.85 (1.05) | 3.83 | 1.22 | 48.7 | 29.2 | 13.0 | 6.1 | 3.0 |
| BE2_thought read | 0 | [0–1] | 0.52 (.89) | 6.70 | 1.96 | 66.8 | 21.0 | 7.7 | 2.8 | 1.8 |
| BE3_tought own | 0 | [0–1] | 0.55 (.90) | 6.26 | 1.86 | 64.4 | 23.0 | 7.7 | 3.2 | 1.8 |
| BE4_thought vivid | 0 | [0–1] | 0.67 (1.05) | 5.22 | 1.70 | 61.6 | 21.6 | 9.0 | 4.0 | 3.8 |
| BE5_thought echo | 0 | [0–1] | 0.75 (1.05) | 4.66 | 1.50 | 55.3 | 26.2 | 10.4 | 4.5 | 3.6 |
| BE6_control external forces | 0 | [0–0] | 0.35 (.79) | 10.77 | 2.72 | 77.5 | 14.6 | 4.4 | 1.8 | 1.7 |
| BE7_double place | 0 | [0–0] | 0.26 (.70) | 14.50 | 3.27 | 83.5 | 10.5 | 3.4 | 1.4 | 1.3 |
| PA1_heard voices | 0 | [0–1] | 0.48 (.87) | 7.56 | 2.15 | 69.9 | 19.1 | 6.6 | 2.5 | 1.9 |
| PA2_heard voices talking | 0 | [0–0] | 0.28 (.69) | 13.37 | 3.09 | 81.9 | 12.2 | 3.3 | 1.7 | 0.9 |
| PA3_seen things | 0 | [0–1] | 0.39 (.80) | 9.39 | 2.48 | 74.3 | 17.3 | 4.6 | 2.3 | 1.4 |

Note: PI = paranoid ideation; BE = bizarre experiences; PA = perceptual anomalies.

**Table 3. Fit indices, unidimensional and hierarchical models, CAPE-P15 scale.**

| Index | Abbreviation | Estimator 1Factor | Estimator Model2 | Good fit | Acceptable fit |
|---|---|---|---|---|---|
| Root mean square error of approximation | RMSEA | 0.09 | 0.04 | < = 0.05 | < = 0.08 |
| Standardized root mean square residual | SRMR | 0.09 | 0.05 | < = 0.1 | < = 0.1 |
| Normedfitindex | NFI | 0.97 | 0.99 | > = 0.95 | > = 0.90 |
| Non-normedfitindex | NNFI | 0.97 | 0.99 | > = 0.97 | > = 0.95 |
| Comparativefitindex CFI | CFI | 0.97 | 0.99 | > = 0.97 | > = 0.95 |
| Goodness of fit index | GFI | 0.98 | 0.99 | >0.95 | >0.90 |
| Adjusted goodness of fit index | AGFI | 0.97 | 0.99 | >0.90 | >0.90 |
| Comparative test between models | Chisq | $\chi^2 (90) = 512.1$ | $\chi^2 (87) = 276.5$ | $\chi^2 (3) = 235.6$ ($p < 0.001$) | |

paranoid ideation (PI1, PI2, and PI5) and auditory hallucinations (PA1), and very high values for items assessing paranoid ideation (PI3 and PI4), bizarre experiences (BE2, BE3, BE4, BE5, BE6, BE7), and auditory hallucinations (PA2).

Concerning the difficulty parameters $\beta_1$ and $\beta_2$, we observed clear differences in PE severity. The means of these two parameters were .36 for $\beta_1$ (range = -.73–1.36) and 1.5 for $\beta_2$ (range = -.23–2.35). Item PI2 showed the lowest latent trait ($\beta_1 = 1.73$; $\beta_2 = -.23$). Therefore, a person located close to the mean of the latent trait (within a region without diagnostic value) will be highly likely to give a positive answer. By contrast, the items with the higher thresholds were BE7 ($\beta_1 = 1.36$; $\beta_2 = 2.21$), PA2 ($\beta_1 = 1.27$; $\beta_2 = 2.25$), and PA3 ($\beta_1 = 1.08$; $\beta_2 = 2.35$), which means that a positive response to them represents a high PE level.

**Item information function.** As depicted in Fig 1, item PI2 provides information at lower psychotic levels. Moreover, item BE1 provides information throughout all ranges of the latent variable, and therefore it appears not to discriminate it properly in this population. By contrast, items BE7, PA2, and PA3 provide the most information at high levels of PE.

**Test information function.** Fig 2 represents the Test Information Function (TIF), which is equivalent to the combined value of the information functions of the fifteen items

**Table 4. Discrimination parameters, CAPE-P15 items.**

| Items | Alpha | $\beta_1$ | $\beta_2$ |
|---|---|---|---|
| PI1_drop hints | 1.55 | -0.59 | 1.03 |
| PI2_ seemto be | 1.36 | -1.73 | -0.23 |
| PI3_persecuted | 1.90 | 0.32 | 1.57 |
| PI4_conspiracy | 1.89 | 0.59 | 1.70 |
| PI5_look oddly | 1.58 | 0.04 | 1.24 |
| BE1_electronic devices | 1.05 | -0.06 | 1.45 |
| BE2_thought read | 1.98 | 0.57 | 1.56 |
| BE3_tought own | 2.08 | 0.45 | 1.47 |
| BE4_thought vivid | 2.11 | 0.36 | 1.23 |
| BE5_thought echo | 1.92 | 0.16 | 1.19 |
| BE6_control externalforces | 2.44 | 0.91 | 1.75 |
| BE7_double place | 1.74 | 1.36 | 2.21 |
| PA1_heard voices | 1.61 | 0.74 | 1.79 |
| PA2_heard voicestalking | 1.72 | 1.27 | 2.25 |
| PA3_seen things | 1.27 | 1.08 | 2.35 |

Note: Discrimination values: 0 = No discrimination; 0.01–0.34 = very low; 0.35–0.64 = low; 0.65–1.34 = moderate; 1.35–1.69 = high; > = 1.7 = very high.

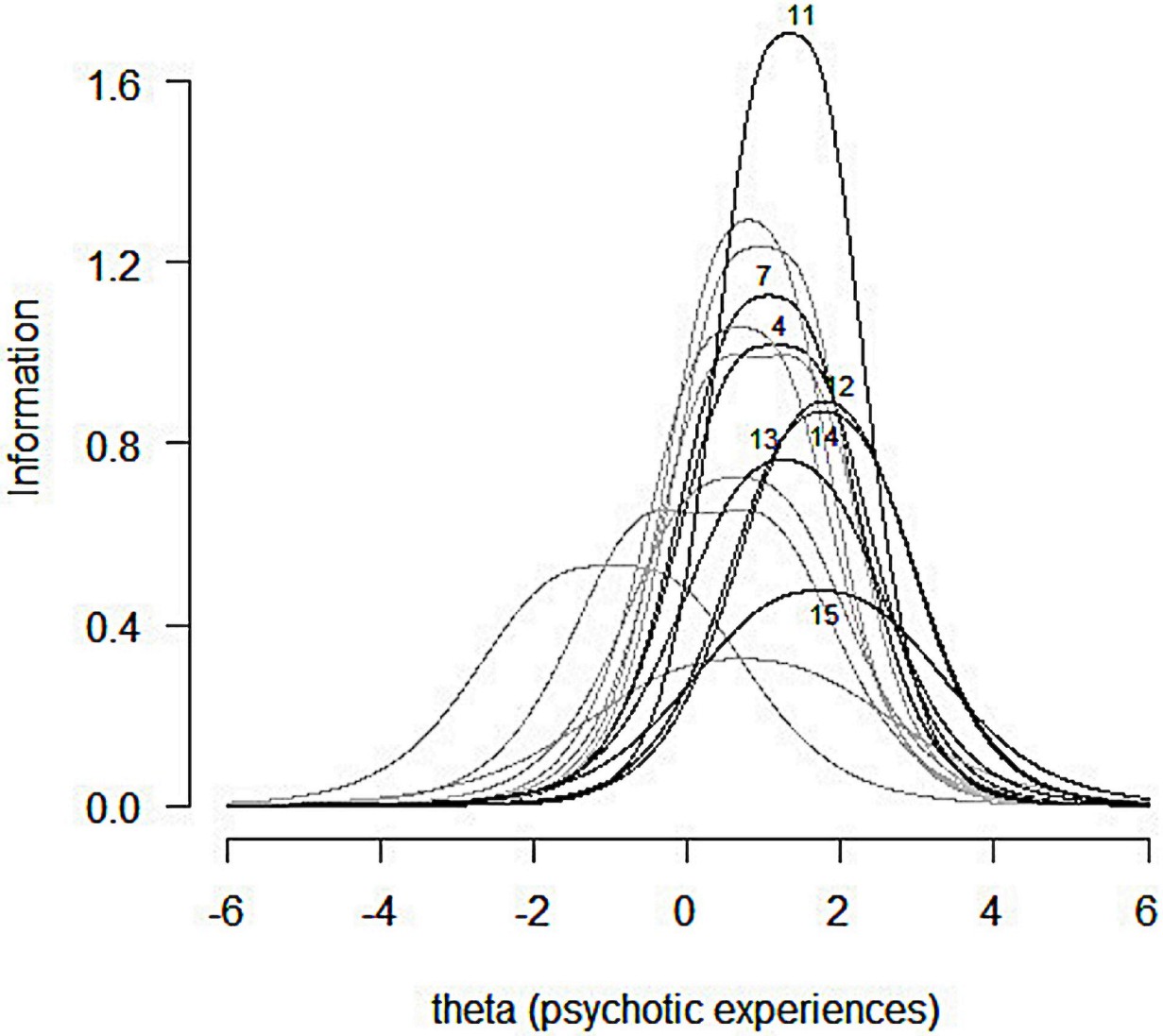

**Fig 1. Item information function.** X-axis represents the severity of PE.

(continuous line), and the distribution of the measurement error (discontinuous line). We observe that the test provides information throughout the entire course of the latent variable, being more precise between -1SD and 2.5 SD.

**Relationship between PE and other variables.** Three groups were defined: Low $\leq$ -0.93 SD; Medium = between -0.93 and 0.93 SD; and High $\geq$ 0.93 SD. We observe significant differences among groups in all variables (Fig 3; S1 Table). Also, higher values for the PE category are associated with higher scores for depressive symptoms, anxiety symptoms, suicidal ideation, rumination, defeat, and entrapment.

## Discussion

To the best of our knowledge, this is the first study examining the accuracy of the CAPE-P15 for differentiating participants according to their PE severity levels in a large sample of adolescent school students aged 12 to 19. We analyzed the difficulty and the discrimination power of

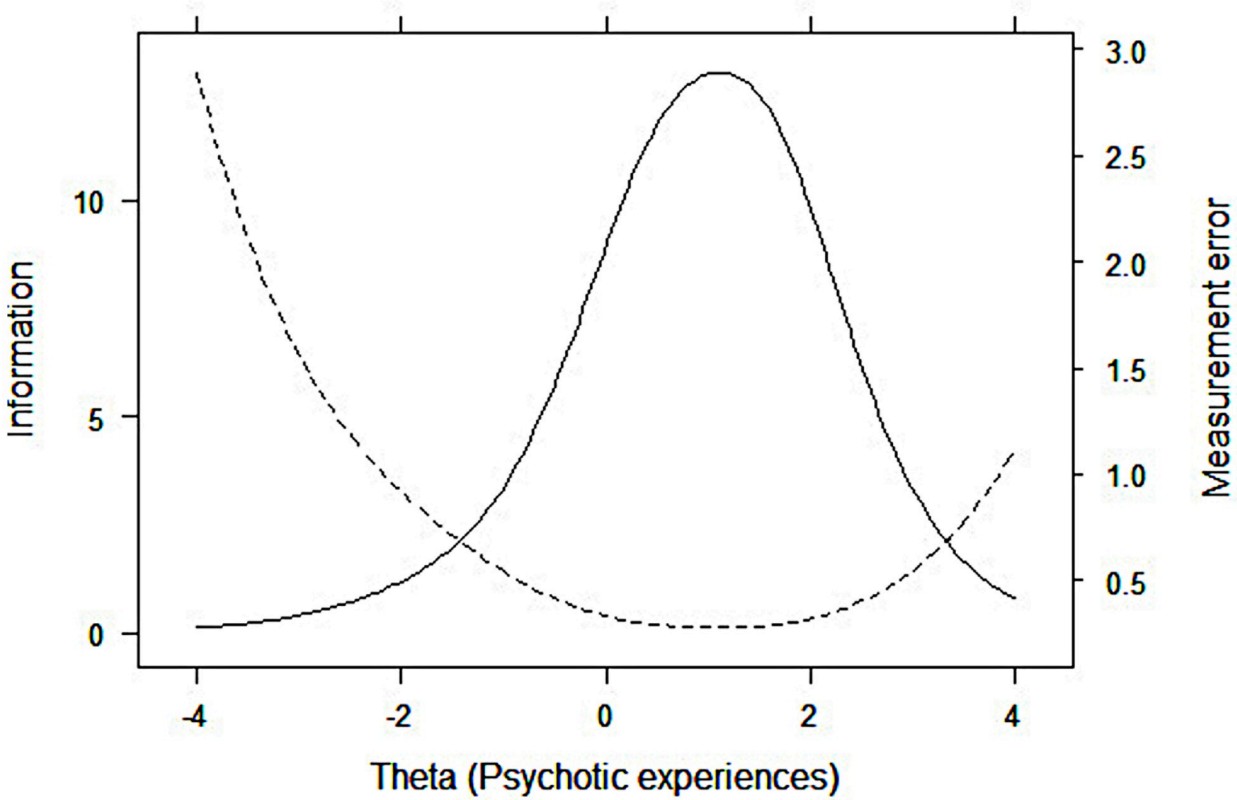

**Fig 2. Test information function (TIF) and measurement error distribution, CAPE-P15 scale.** The solid line represents the TIF, which is equivalent to the combined value of the information functions of the fifteen items of the CAPE-P15. The dotted line represents the standard error.

each item as well as the accuracy with which the scale provides information about the latent trait. The capability of the items to discriminate PE ranged from moderate to very high. The item information function analysis yielded a high ability for the items to differentiate severity levels of PE. The test information function analysis showed that the scale provides very accurate information about PE, particularly for high PE levels. Our findings support prior research showing that the CAPE-P15 is a suitable tool for screening PE in the general population [31,35] and provide new evidence on the accuracy of each item in classifying and differentiating PE levels in school-age adolescents. These results, owing to the inclusion of PE as a meaningful domain [1], are relevant in the context of the increasing interest in conducting research on psychosis in general populations [24] and the well-recognized need to detect individuals at risk for mental health problems.

The suitability of the CAPE-P15 for screening PE in the general population is additionally supported by its reliability, which was good in the present study. Our results mirror previous research showing satisfactory reliability in secondary students [34], university and college students [32,35], primary care [57], and mental health services [3]. Our findings on the internal structure of the CAPE-P15 revealed acceptable fit values for both the unidimensional and the hierarchical models, with a slightly better fit for the latter. This is consistent with prior research suggesting the usage of a mean total score [3,35]. However, given prior evidence supporting a three-factor structure [32–34], combining general and specific dimensions of PE could be informative for diagnostic purposes [58]. Additionally, using subtypes of PE is currently recommended [24], for instance, considering their potential specific role in the formation of

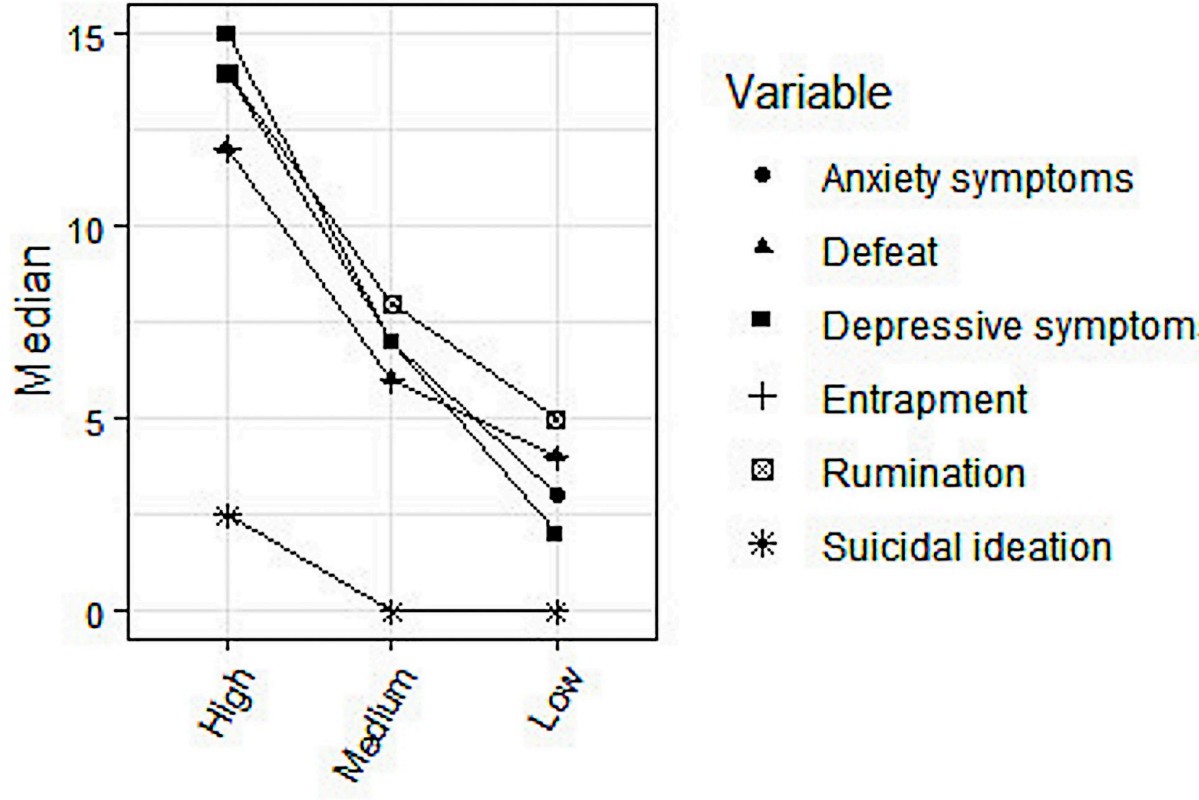

**Fig 3. Associations between severity of PE and depression/anxiety symptoms, suicidal ideation, defeat, entrapment, and rumination.**

psychopathology (e.g. suspiciousness strongly associated with vulnerability to psychosis; ideas of reference more strongly linked to development of affective disorders [59]).

Regarding the general functioning of the scale, we observed a floor effect, meaning that most of the participants obtained low PE scores. The scale is productive within a small range of the variable, particularly in moderate-high and very high areas, which is to be expected for clinical scales administered to a general population. Most items showed a high discriminative capability [56], especially those assessing delusional experiences of being controlled by external forces (BE6) and thought control/broadcasting (items BE2, BE3, BE4, and BE5). The item "Have you ever felt as if some people are not what they seem to be?" (PI2) reached the highest mean value, indicating that it is the most frequent PE in our sample, which mirrors other findings [35]. Although its discriminative ability (the capability to differentiate between levels of the latent trait) was high, it showed the lowest capability to determine the presence of PE within the higher levels of the trait. By contrast, this capability was very high for the items with lower mean values, which assessed perceptual anomalies (auditory, PA1 and PA2; visual hallucinations, PA3), bizarre experiences (BE7 = a double has taken the place of a family member, friend or acquaintance; BE6 = being controlled by external forces), and persecutory ideation (PI4 = conspiracy against me). Thus, as expected, we observed a general pattern where the more weakly a symptom was experienced by participants the higher was its capability to determine the presence of the latent trait within the higher severity levels. This supports prior research showing that questionnaires assessing PE provide limited and less reliable information at lower trait levels [60].

According to both severity and peak information, the items can be categorized into three groups with different levels of PE. Group 1 (low presence) encompasses paranoid ideation (PI2); group 2 (intermediate presence) includes items from PI (PI1, PI3, and PI5) and bizarre experiences (BE1, BE3, BE4, and BE5); and group 3 (high presence) comprises items from PI (PI4), BE (BE2, BE6, and BE7), and perceptual anomalies (PA1, PA2, and PA3). Concerning item BE1 ("Have you ever felt as if electrical devices such as computers can influence the way you think?"), we observed an unclear pattern: whereas the difficulty parameter β2 shows a high capability to differentiate the latent trait, the item information function does not support this assertion. Results reported by Kelleher et al. [61] and ours, though using different item wording for similar content ("Have you ever had messages sent just to you through TV or radio?" vs. "Have you ever felt as if electrical devices such as computers can influence the way you think?"), showed the poorest predictive capability. Regarding item 6, one potential explanation for this result may be related to the generalized use of social media or internet activity in the present day. Currently, the item wording probably does not capture the psychopathological nature of this phenomenon. However, this is assertion needs further research.

Our results support prior research showing that visual and auditory hallucinations, when assessed with the Prodromal Questionnaire-Brief Child Version (PQ-BC) [62], discriminate the latent trait in a sample of children aged 9 to 10 years [60]. Moreover, our findings are also in accordance with Phalen et al. [63], who, in help-seeking adolescents and college participants, observed that these two dimensions were the most informative of PE when assessed using three psychosis screening tools: the Prime Screen [64], the Prodromal Questionnaire-Brief (PQ-B) [62], and the Youth Psychosis at Risk Questionnaire-Brief (YPARQ-B) [65]. Our results also mirror the findings reported by Kelleher et al. [61], who found that these symptoms plus paranoid delusional experiences and feelings of being controlled by external forces were highly specific and sensitive for detecting PE in adolescents. However, because different questionnaires and age ranges were used, direct comparisons should be interpreted with caution. Importantly, our most informative seven items belong to our Group 3, comprising PE domains previously reported as being strongly associated with distress, depression, and poor functioning [66]. Further research examining whether specific items are differentially associated with psychological difficulties and with different psychopathological outcomes is needed and strongly encouraged nowadays [24,59,67].

Finally, we observed significant between-group differences reflecting strong positive associations between the intensity of PE and emotional distress measured via emotional (depressive and anxiety) symptoms, suicidal ideation, and maladaptive psychological factors or cognitive distortions previously associated with depression and suicide (defeat, entrapment, and rumination) [40,41]. This indicates good discriminant validity and is in line with prior research showing that, in young people from the general population, PE can be regarded as risk markers for a wide range of non-psychotic symptoms [6,68], emotional distress, and suicidal ideation [9,69–72]. Additionally, our findings revealing associations between PE and cognitive distortions support prior research conducted in the light of cognitive models of psychosis [73,74]. For instance, rumination has been associated with negative symptoms [75] and has been found to act as a predictor of persecutory delusional and hallucinatory experiences [38,76]. Nevertheless, it is not clear whether ruminative processing is a reaction to psychotic symptoms or a true precursor [39]. Moreover, entrapment has been suggested to mediate the association between the severity of positive symptoms (particularly suspiciousness) and suicidal ideation in schizophrenia patients [41]. As reported by Valmaggia et al. [77], defeat and entrapment contribute to the onset of psychopathology in people at risk for psychosis, probably triggering paranoid appraisals in social contexts. This association could be mediated by attenuated psychotic symptoms, which deserves further research. However, testing the impact and specific

mechanisms underlying associations between PE, psychiatric symptoms, suicidal ideation, and cognitive distortions is beyond the scope of the present study. Future research examining these issues could provide new insights to improve preventive interventions aimed at enhancing protective coping strategies in those at risk for mental health problems.

This study has some limitations. First, given our cross-sectional design, we cannot establish causal relationships among variables. Second, we recruited participants from public schools. Therefore, adolescents from higher socioeconomic status backgrounds, usually attending private schools, could be underrepresented. Third, despite the adequate validity of the CAPE-P15 for assessing PE and the adequate sensitivity and specificity of self-report measures for identifying PE in adolescents, using a self-report questionnaire might over-estimate the prevalence of these experiences [10]. However, the endorsement rates of PE were similar to those of recent studies conducted with relatively similar-aged samples [6,35]. Fourth, we assessed rumination using a limited set of items, and probably failed to capture some specific aspects of this construct. Fifth, because we did not address the distress associated with PE, we cannot draw conclusions regarding the clinical relevance of PE in our sample. Sixth, we did not use exclusion criteria for participation in the study. Finally, because the IRT model used assumes one-dimensionality, we did not compute between-group comparisons to assess associations between subtypes of PE and other variables. A hierarchical latent IRT model considering a hierarchical structure should be tested if feasible.

## Conclusions

Based on item response theory, our results demonstrate that the CAPE-P15 is a reliable and useful tool for screening PE in adolescents from the general population and for accurately classifying them according to their differential levels of PE.

## Supporting information

**S1 Fig. Unidimensional and hierarchical models.** A) General factor mode. B) Hierarchical model.
(TIF)

**S1 Table. Relationship between PE and other variables.**
(XLSX)

## Acknowledgments

The authors would like to acknowledge the important role of all participants, both students and professionals, who supported the project.

## Author Contributions

**Conceptualization:** D. Núñez, J. Gaete.

**Formal analysis:** M. I. Godoy.

**Funding acquisition:** D. Núñez.

**Investigation:** D. Núñez, M. J. Faúndez, S. Campos, A. Fresno, R. Spencer.

**Project administration:** D. Núñez, R. Spencer.

**Writing – review & editing:** D. Núñez, J. Gaete.

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
