## [Decision Letter · Decision Letter 0]

30 May 2021

PONE-D-20-34868

The Community Assessment of Psychic Experiences-Positive scale (CAPE-P15) accurately classifies and differentiates psychotic experiences levels in adolescents from the general population

PLOS ONE

Dear Dr. Núñez,

Thank you for submitting your manuscript to PLOS ONE. After careful consideration, we feel that it has merit but does not fully meet PLOS ONE’s publication criteria as it currently stands. Therefore, we invite you to submit a revised version of the manuscript that addresses the points raised during the review process.

We look forward to receiving your revised manuscript.

Kind regards,

Christine Mohr, PhD

Academic Editor

PLOS ONE

Journal Requirements:

2. Please include a copy of Table 5 which you refer to in your text on page 13.

Additional Editor Comments (if provided):

First, I would like to apologize for the time it took to find suitable persons to review your manuscript. Yet, once found, you can read that both reviewers appreciate your manuscript, and have only minor comments, on how to further improve your manuscript.

The comments are precise, and well explained.

Please be careful with regards to clarity and text editing, consistency of use of terms, and labels.

Being myself not a "native speaker", we always profit from a native speaker going over the text

Reviewers' comments:

Reviewer's Responses to Questions

**Comments to the Author**

1. Is the manuscript technically sound, and do the data support the conclusions?

Reviewer #1: Yes

Reviewer #2: Yes

2. Has the statistical analysis been performed appropriately and rigorously? 

Reviewer #1: Yes

Reviewer #2: Yes

3. Have the authors made all data underlying the findings in their manuscript fully available?

Reviewer #1: Yes

Reviewer #2: Yes

4. Is the manuscript presented in an intelligible fashion and written in standard English?

Reviewer #1: Yes

Reviewer #2: Yes

5. Review Comments to the Author

Reviewer #1: “The Community Assessment of Psychic Experiences-Positive scale (CAPE-P15) accurately classifies and differentiates psychotic experiences levels in adolescents from the general population”

In this study, the authors applied IRT in order to examine the psychometric properties of CAPE-P15 and the identification of psychotic experiences in an adolescent sample. The study is interesting, it is of scientific and clinical interest and I only have a few minor comments prior to publication.

1. The authors report the inclusion but not the exclusion criteria for participation. Were any participants excluded due to alcohol/drug abuse, parents diagnosed with psychosis or for any other reason? If not, I suggest the authors include this in the limitations of the study. If yes, please report, as appropriately.

2. In the description of the scale in the methods section, it would be useful to include a brief description of the sub-scales (this could help the readers who are not familiar with the instrument).

3. Please replace “subjects” with participants/individuals throughout the manuscript.

4. Please spell out PE at first mention in the abstract.

5. The sentence beginning “Another approach could be to establish…” in the introduction requires re-phrasing.

6. I think that “…should be cautionary interpreted” in the discussion, should be re-phrased into “… interpreted with caution”.

Reviewer #2: The Community Assessment of Psychic Experiences-Positive scale accurately classifies and differentiates psychotic experiences levels in adolescents from the general population.

The study assesses the psychometric properties of the CAPE-P15 scale in a sample of adolescents from the general population. The overall findings were positive indicating utility within this population. The authors demonstrated that the task

This article is informative and would benefit the field. As the authors noted, there is a growing view of psychotic symptoms existing on a spectrum, and an interest in detecting subclinical levels within the general population. Validating scales such as the CAPE-P15 within healthy or nonclinical populations will be necessary to build our understanding of the impact these experiences have across the spectrum.

The research conducted seems sound and should be disseminated. However, I recommend revisions to improve the clarity of the manuscript. Overall, I believe that the manuscript would benefit from copy editing as there were several errors in subject-verb agreement or incongruous verb tenses that decreased the readability of the manuscript. These do not prevent the reader from understanding the overall results, but it does obscure the message and may make it difficult for those unfamiliar with the task to determine the impact of the manuscript.

Below are specific comments on clarifications and suggested edits.

1. For all scales used, it would be beneficial to include a scoring criterion (e.g. all items are summed for a range of # to #, with higher scores indicating more of the trait being assessed, or all items are averaged for an overall measure of the trait being assessed).

2. In discussing specific items, it would be best to use consistent naming criteria throughout the manuscript, and its preferred that you utilize an informative item naming system (unless you are able to include a copy of the assessment within supplemental materials). In table 2, you use a subdomain abbreviation with item number – along with a brief description. I would utilize this naming system throughout, and interested researchers can look back to this table to match the named item to the item content.

3. Section 3.2 could be expanded. What does it mean about the scale that the model with three correlated factors performed better to the reader? (The authors do this well in the following section 3.3).

4. The authors note that their Item 6 did not discriminate as it did in previous research due to a difference in phrasing of the item. In reading the characterization of the item, I can see how different “electrical devices influence the way we think” is from someone receiving messages “sent just to you” especially how much media attention has been devoted to examining the impact of social media or internet activity has on the way we think. The question I have is how do the authors account for this discrepant message? Are there any other meaningful differences between the version used for this research and the versions used in alternate studies cited?

Minor comments:

1. In the abstract, SI abbreviation is used without indicating that this refers to suicidal ideation prior.

2. In the introduction, a cut-off score for the CAPE is discussed without referencing the scoring criteria prior.

3. In methods/results, the name for McDonald’s ω is inconsistently used. If the authors do not wish to use the name throughout the document, introduce it at the first instance (same with Chronbach’s alpha).

4. It may be beneficial to give a brief description for defeat and entrapment as these are less commonly used terms.

5. On page 20, the term “scholar adolescents” is used, but it is unclear to what this term is referring? Is this just that all adolescents were current students?

6. PLOS authors have the option to publish the peer review history of their article (what does this mean?). If published, this will include your full peer review and any attached files.

Reviewer #1: No

Reviewer #2: No

---

## [Author Response · Author response to Decision Letter 0]

12 Jul 2021

We would like to thank to the reviewers for the helpful comments that enabled us to improve the quality of the manuscript. All issues raised by the reviewers are highlighted in the manuscript and are addressed point by point below.

Reviewer #1: 

1. The authors report the inclusion but not the exclusion criteria for participation. Were any participants excluded due to alcohol/drug abuse, parents diagnosed with psychosis or for any other reason? If not, I suggest the authors include this in the limitations of the study. If yes, please report, as appropriately.

Reply: We included the point raised by the reviewer as a limitation of the study: “We did not use exclusion criteria for participation in the study”. 

2. In the description of the scale in the methods section, it would be useful to include a brief description of the sub-scales (this could help the readers who are not familiar with the instrument).

Reply: We included the following description of the subscales: “The scale assesses three domains: Paranoid ideation (PI, 5 items), Bizarre experiences (BE, 7 items), and perceptual anomalies (PA, 3 items). Higher scores indicate higher severity of PE. All items are averaged for an overall measure of the trait being assessed”.

3. Please replace “subjects” with participants/individuals throughout the manuscript.

Reply: We replaced “subjects” with individuals/participants (p. 3, 16).

4. Please spell out PE at first mention in the abstract.

Reply: We did spell out PE as suggested. 

5. The sentence beginning “Another approach could be to establish…” in the introduction requires re-phrasing.

Reply: The sentence was re-phrased. Now it says: “Alternatively, it would be possible to establish several risk groups according to the severity of the latent traits using scores derived from Item Response Theory analysis”

6. I think that “…should be cautionary interpreted” in the discussion, should be re-phrased into “… interpreted with caution”.

Reply: The sentence was changed as suggested by the reviewer

Reviewer #2: The Community Assessment of Psychic Experiences-Positive scale accurately classifies and differentiates psychotic experiences levels in adolescents from the general population.

1. For all scales used, it would be beneficial to include a scoring criterion (e.g. all items are summed for a range of # to #, with higher scores indicating more of the trait being assessed, or all items are averaged for an overall measure of the trait being assessed).

Reply: We have included scoring criteria for all scales used. 

2. In discussing specific items, it would be best to use consistent naming criteria throughout the manuscript, and its preferred that you utilize an informative item naming system (unless you are able to include a copy of the assessment within supplemental materials). In table 2, you use a subdomain abbreviation with item number – along with a brief description. I would utilize this naming system throughout, and interested researchers can look back to this table to match the named item to the item content.

Reply: We changed the names of the items in the text, and now we are using the naming system suggested by the reviewer. 

3. Section 3.2 could be expanded. What does it mean about the scale that the model with three correlated factors performed better to the reader? (The authors do this well in the following section 3.3).

Reply: We modified the section as follows. “The fit indices of the CAPE-P15 are good (Table 3). The unidimensional structure of the CAPE (Estimator 1 Factor= 1 general factor reflecting an average score of all items) was corroborated. The hierarchical structure (Estimator Model 2= 1 general factor plus three correlated factors) was also corroborated. The RMESA index was acceptable for both structures but slightly better for the hierarchical structure (Supplementary Figures 1a and 1b). Thus, the data support the existence of a general PE latent factor, but at the same time, it is possible to differentiate three specific PE dimensions which may have different clinical meanings. 

4. The authors note that their Item 6 did not discriminate as it did in previous research due to a difference in phrasing of the item. In reading the characterization of the item, I can see how different “electrical devices influence the way we think” is from someone receiving messages “sent just to you” especially how much media attention has been devoted to examining the impact of social media or internet activity has on the way we think. The question I have is how do the authors account for this discrepant message? Are there any other meaningful differences between the version used for this research and the versions used in alternate studies cited?

Reply: In relation to the first question, we corrected this observation regarding the usefulness of item 6 to discriminate symptoms among adolescents. Results reported by Kelleher et al. [61] and ours, though using different item wording for similar content (“Have you ever had messages sent just to you through TV or radio?” vs “Have you ever felt as if electrical devices such as computers can influence the way you think?”), showed the poorest predictive capability. Regarding item 6, one potential explanation for this result may be related to the generalized use of social media or internet activity in the present day. Currently, the item wording probably does not capture the psychopathological nature of this phenomenon. However, this is assertion needs further research”.

Regarding the question on meaningful differences between the version used for this research and the versions used in alternate studies, we have included other comparisons and modified the manuscript as follows: “Our results support prior research showing that visual and auditory hallucinations, when assessed with the Prodromal Questionnaire-Brief Child Version (PQ-BC) [62], discriminate the latent trait in a sample of children aged 9 to 10 years [60]. Moreover, our findings are also in accordance with Phalen et al. [63], who, in help-seeking adolescents and college participants, observed that these two dimensions were the most informative of PE when assessed using three psychosis screening tools: the Prime Screen [64], the Prodromal Questionnaire-Brief (PQ-B) [62], and the Youth Psychosis at Risk Questionnaire-Brief (YPARQ-B)” [65].

References

Kelleher I, Wigman JTW, Harley M, O’Hanblon E. Psychotic experiences in the population: Association with functioning and mental distress. Schizophrenia Research. 2015;165: 9-14.

Loewy, R. L., Pearson, R., Vinogradov, S., Bearden, C. E., & Cannon, T. D. (2011). Psychosis risk screening with the Prodromal Questionnaire—Brief version (PQ-B). Schizophrenia Research, 129, 42–46. http://dx.doi.org/10.1016/j.schres.2011.03.029

Miller, T.J., Cicchetti, D., Markovich, P.J., McGlashan, T.H., Woods, S.W., 2004. The SIPS screen: a brief self-report screen to detect the schizophrenia prodrome. Schizophr. Res. 70 (1), 78.

Ord, L., Myles-Worsley, M., Blailes, F., Ngiralmau, H., 2004. Screening for prodromal adolescents in an isolated high-risk population. Schizophr. Res. 71 (2–3), 507–508.

Minor comments:

1. In the abstract, SI abbreviation is used without indicating that this refers to suicidal ideation prior.

Reply: We included “suicidal ideation” before SI.

2. In the introduction, a cut-off score for the CAPE is discussed without referencing the scoring criteria prior.

Reply: We clarified that the scoring criterion was the average of the scale’s items: “For example, Bukenaite et al. [3], computing the average of the items, identified a cut-off of 1.47….”. 

3. In methods/results, the name for McDonald’s ω is inconsistently used. If the authors do not wish to use the name throughout the document, introduce it at the first instance (same with Chronbach’s alpha).

Reply: We have changed the names according to the reviewer suggestion.

4. It may be beneficial to give a brief description for defeat and entrapment as these are less commonly used terms.

Reply: We defined these constructs when describing the scale assessing them.

5. On page 20, the term “scholar adolescents” is used, but it is unclear to what this term is referring? Is this just that all adolescents were current students?

Reply: We replaced “scholar adolescents” by “adolescent school students”.

---

## [Decision Letter · Decision Letter 1]

13 Aug 2021

The Community Assessment of Psychic Experiences-Positive scale (CAPE-P15) accurately classifies and differentiates psychotic experiences levels in adolescents from the general population

PONE-D-20-34868R1

Dear Dr. Núñez,

We’re pleased to inform you that your manuscript has been judged scientifically suitable for publication and will be formally accepted for publication once it meets all outstanding technical requirements.

Kind regards,

Christine Mohr, PhD

Academic Editor

PLOS ONE

Additional Editor Comments:

Reviewers' comments:

Reviewer's Responses to Questions

**Comments to the Author**

1. If the authors have adequately addressed your comments raised in a previous round of review and you feel that this manuscript is now acceptable for publication, you may indicate that here to bypass the “Comments to the Author” section, enter your conflict of interest statement in the “Confidential to Editor” section, and submit your "Accept" recommendation.

Reviewer #1: All comments have been addressed

Reviewer #2: All comments have been addressed

2. Is the manuscript technically sound, and do the data support the conclusions?

Reviewer #1: Yes

Reviewer #2: Yes

3. Has the statistical analysis been performed appropriately and rigorously? 

Reviewer #1: Yes

Reviewer #2: Yes

4. Have the authors made all data underlying the findings in their manuscript fully available?

Reviewer #1: Yes

Reviewer #2: Yes

5. Is the manuscript presented in an intelligible fashion and written in standard English?

Reviewer #1: Yes

Reviewer #2: Yes

6. Review Comments to the Author

Reviewer #1: (No Response)

Reviewer #2: I appreciate the attention the authors dedicated to the reviewer's comments. All queries have been addressed and the manuscript is in excellent condition for acceptance.

7. PLOS authors have the option to publish the peer review history of their article (what does this mean?). If published, this will include your full peer review and any attached files.

Reviewer #1: No

Reviewer #2: **Yes: **Hans S. Klein

---

## [Editor Report · Acceptance letter]

18 Aug 2021

PONE-D-20-34868R1 

The Community Assessment of Psychic Experiences-Positive scale (CAPE-P15) accurately classifies and differentiates psychotic experience levels in adolescents from the general population 

Dear Dr. Núñez:

I'm pleased to inform you that your manuscript has been deemed suitable for publication in PLOS ONE. Congratulations! Your manuscript is now with our production department. 

Kind regards, 

on behalf of

Dr. Christine Mohr 

Academic Editor

PLOS ONE